# Conductive Carbon Materials from the Hydrothermal Carbonization of Vineyard Residues for the Application in Electrochemical Double-Layer Capacitors (EDLCs) and Direct Carbon Fuel Cells (DCFCs)

**DOI:** 10.3390/ma12101703

**Published:** 2019-05-26

**Authors:** Viola Hoffmann, Dennis Jung, Joscha Zimmermann, Catalina Rodriguez Correa, Amal Elleuch, Kamel Halouani, Andrea Kruse

**Affiliations:** 1Department of Conversion Technologies of Biobased Resources, Institute of Agricultural Engineering, University of Hohenheim, Garbenstrasse 9, 70599 Stuttgart, Germany; dennis.jung@uni-hohenheim.de (D.J.); c.rodriguez@uni-hohenheim.de (C.R.C.); andrea_kruse@uni-hohenheim.de (A.K.); 2Institute of Catalysis Research and Technology (IKFT), Karlsruhe Institute for Technology (KIT), Hermann-von-Helmholtz-Platz 1, 76344 Eggenstein-Leopoldshafen, Germany; joscha.zimmermann@kit.edu; 3National Engineering School of Sfax, University of Sfax, Micro Electro Thermal Systems (UR13ES76), IPEIS, Road Menzel Chaker km 0.5 P.O. Box 1172, 3018 Sfax, Tunisia; amal-elleuch@hotmail.com (A.E.); kamel_ipeis@yahoo.fr (K.H.); 4Digital Research Center of Sfax, Technopole of Sfax, P.O. Box 275, Sakiet Ezzit, 3021 Sfax, Tunisia

**Keywords:** hydrothermal carbonization, grape pomace, vine pruning, bio-based carbon materials, electrical conductivity, supercapacitor, direct carbon fuel cell, pyrolysis, energy storage, advanced carbon materials

## Abstract

This study investigates the production of bio-based carbon materials for energy storage and conversion devices based on two different vineyard residues (pruning, pomace) and cellulose as a model biomass. Three different char categories were produced via pyrolysis at 900 °C for 2 h (biochars, BC), hydrothermal carbonization (HTC) (at 220, 240 or 260 °C) with different reaction times (60, 120 or 300 min) (hydrochars, HC), or HTC plus pyrolysis (pyrolyzed hydrochars, PHC). Physicochemical, structural, and electrical properties of the chars were assessed by elemental and proximate analysis, gas adsorption surface analysis with N_2_ and CO_2_, compression ratio, bulk density, and electrical conductivity (EC) measurements. Thermogravimetric analysis allowed conclusions to be made about the thermochemical conversion processes. Taking into consideration the required material properties for the application in electrochemical double-layer capacitors (EDLC) or in a direct carbon fuel cell (DCFC), the suitability of the obtained materials for each application is discussed. Promising materials with surface areas up to 711 m^2^ g^−1^ and presence of microporosity have been produced. It is shown that HTC plus pyrolysis from cellulose and pruning leads to better properties regarding aromatic carbon structures, carbon content (>90 wt.%), EC (up to 179 S m^−1^), and porosity compared to one-step treatments, resulting in suitable materials for an EDLC application. The one-step pyrolysis process and the resulting chars with lower carbon contents and low EC values between 51 and 56 S m^−1^ are preferred for DCFC applications. To conclude, biomass potentials can be exploited by producing tailored biomass-derived carbon materials via different carbonization processes for a wide range of applications in the field of energy storage and conversion.

## 1. Introduction

Energy storage and conversion technologies, such as Electrochemical Double-Layer Capacitors (EDLC) or Fuel Cells (FC), are key technologies in the context of electro mobility and the sustainable energy sector. While EDLCs and certain fuel cells, such as hydrogen fuel cells (HFC), have already found broad fields of applications, e.g., in hybrid drive technology [1] or regenerative braking systems [2], direct carbon fuel cells (DCFCs) are still in the developmental stage [3]. Especially regarding the charge-transfer processes in a DCFC, a lot of fundamental research is still required before a practical implementation of the technology becomes realistic [4]. Nevertheless, compared to other fuel cells based on gaseous or liquid fuels, DCFCs use a solid fuel (carbon) to directly convert their chemical energy into electrical energy, resulting in a phase separation of fuel and product (see overall fuel cell reactions in Equations (1) and (2) [5]. This creates additional challenges, such as the feeding of the solid carbon into the cell.

O_2_ (air) + 4e = 2O^2−^(1)

C + 2O^2−^ = CO_2_ + 4e(2)

Since the entropy term (∆S) of the carbon oxidation reaction is close to zero and its thermodynamic efficiency is almost 100% (temperature-independent), as a consequence the fuel utilization is also close to 100% (compared to 80–85% for most of other fuel cell types). Considering system-based losses, such as voltage efficiency losses, the stack efficiency of a DCFC system (overall system efficiency) still ranges between 80–85% and higher. This means the electric efficiency is twice as high as for conventional coal-fired power plants, and consequently half as much CO_2_ emissions are produced [3]. However, as already mentioned above, it remains unclear, to a certain extent, which carbon properties lead to the maximum overall efficiency and the reasons behind this. Possible fuels include coal, coke, tar, and even organic waste [6,7,8], but it has been reported that the carbon quality and certain carbon structures are fundamentally important for good long term cell performances and high power densities [9,10]. Some of the required properties are [10]:submicron size carbon particlescrystallographic structure of the carbon latticeelectric conductivity (EC)high specific surface area (SSA)no or low content of impurities (e.g., inorganic matter)/high C content

Based on the constructive characteristics of each DCFC (e.g., type of electrolyte, anode material, fuel delivery method), the named carbon properties can vary slightly but are always aimed at creating the best conditions for the direct oxidation reaction at the electrode/electrolyte interface. In order to not exceed the framework of this paper, an overview of the different DCFC types (electrolytes, etc.) shall not be included here, but it is referred to in the work of Cao et al., 2006 [5], or Giddey and South, 2010 [10].

The above-mentioned carbon properties correspond, to a certain extent, with the required properties for carbonaceous electrode materials in EDLCs [11,12]. For instance, a high SSA increases the performance of EDLCs, due to the electrostatic energy storage mechanism in the electric double-layer on the carbon surface [13,14] according to Equation (3):(3)$$C_{\mathrm{dl}}\;=\frac{\varepsilon A}{4\pi t}$$
where the double-layer capacitance C_dl_ is calculated by multiplying the SSA of the electrode A with the dielectric constant of the double-layer ԑ and dividing it by a multiple of the distance between the double layers, expressed by t. Hence, t expresses the extremely small distance (in Angstroms) between the contrary charged layers on the carbon surface, which is why very high C_dl_ values (16–50 µF·cm^−2^) can be reached in combination with high SSAs [15]. The capacity directly determines the energy density E defined by the formula in Equation (4), whereas the power density of an EDLC (P_max_) is strongly influenced by the inner resistance of the device (see Equation (5)), which in turn is determined by the EC of the electrode material [14].
(4)$$E\;=\;\frac{1}{2}\;C\;V²$$
(5)$$\mathrm{Pmax}\;=\;\frac{V²}{4R}$$

V represents the nominal voltage and R the equivalent series resistance (ESR) in Ω, which decreases with higher EC values of the carbon material. Furthermore, it has been shown that microporous structures are correlated with higher capacities [16], as well as N- and O-containing surface functionalities due to pseudo-capacitive contributions [17,18,19]. High C contents and crystallographic structures of the carbon structure are also favourable for high capacities, because they lead to higher EC values of the respective carbon [20,21].

The carbonization of organic waste via hydrothermal carbonization (HTC) and pyrolysis leads to interesting carbonaceous powder materials, which possess the required characteristics to a greater or lesser extent, depending on the carbonization parameters and conditions. By varying temperature or reaction time of the carbonization process, the carbon properties can even be exactly tailored to the desired application. This makes the biomass carbonization an interesting and promising process for the production of environmentally friendly and bio-based electrode materials, for application in energy storage and conversion technologies, and at the same time, a powerful instrument to convert organic waste streams into value-added products in the context of bioeconomy.

One possible organic waste stream for the conversion into bio-based electrode materials are vineyard residues, which arise in huge amounts during viticulture. In 2016 alone, grape cultivation for vine production amounted to 75 million tons of grape worldwide, associated with large quantities of process wastes, such as grape pomace or vine pruning [22]. Due to their lignocellulosic character and the high moisture content (e.g., >60 wt.% for grape pomace), those biomasses are interesting precursors for thermochemical and especially for hydrothermal processes, such as HTC. However, vineyard pruning can be regarded as a typical lignocellulose biomass, whereas grape pomace has a distinct composition, having a large fraction of condensed tannins, as well as monomeric sugars [23].

This work focusses on the question of if suitable electrode materials with the mentioned properties required for the application in EDLCs and DCFCs can be produced via HTC or pyrolysis of grape pomace and vine prunings. In order to be able to discuss this question properly, the two biomasses and one model biomass (cellulose) were hydrothermally carbonized at different temperatures and reaction times (HTC). After HTC, a certain amount of each obtained hydrochar was pyrolized at 900 °C. Furthermore, the real biomasses and the model substance cellulose were pyrolized at 900 °C without any HTC treatment beforehand. The obtained carbonaceous powders were characterized regarding their physicochemical properties and their electric conductivity (EC) in order to be able to assess their suitability as electrode (anode) materials in EDLCs or fuel materials in DCFCs. For the latter application, the electrical conductivity (EC) has to be discussed, taking into account a certain inconsistency in the literature due to complex interdependencies. On the one hand, the electrical conductivity of carbon materials used as anode materials in DCFCs determine, to a major extent, the global ohmic resistance of a DCFC (high electric conductivity = low ohmic resistance = higher efficiencies) and high EC values are favored by graphitic or crystalline structures of the carbon material. On the other hand, it is often stated in literature that the reactivity of a carbon, which is known to be higher in non-graphitic structures, is more important than a high EC value [24]. Finally, the importance of the different carbon properties highly depends on the DCFC design, which is considered in a specific case. For example, in a molten carbonate fuel cell (MCFC) or in a solid oxide fuel cell (SOFC), the carbon fuel is often also used as the anode electrode, and hence the carbon properties (in turn, determined by their preparation techniques) are of great importance for the cell performance. However, in the case of liquid electrolytes with dispersed carbon particles, carbon properties, such as *EC*, are of minor importance [24]. In this work, the focus will be on MCFCs (with molten carbonate as electrolyte) and SOFCs (solid oxide/ceramic electrolyte) (in the following referred to as DCFCs, since both types are high-temperature DCFCs with different cell designs), since these are the cell designs which are the best studied and understood to date [24]. Nevertheless, most of the fuel types used in DCFCs so far have been commercial activated carbons, carbon blacks, or biomass or waste derived chars [4,24]. In some specific cases, pure biomass has also been tested as a fuel in DCFCs (e.g., waste coffee grounds, [8]). However, the underlying processes and mechanisms or the impact of specific carbon properties, such as EC (or crystallinity), are still not well understood [25]. We attempt to address this knowledge gap in this work by discussing the properties of different biomass-derived carbons with regard to their application in DCFCs. Next to this main research question regarding the suitability of the obtained chars as electrode or solid fuel materials, especially, three more aspects are discussed in the framework of this work: (1) Comparing the properties of biochars, hydrochars, and pyrolyzed hydrochars, how do they differ? (2) Is it possible to transfer results obtained from a model substance (cellulose) to real biomasses? (3) Which correlations can be observed and defined between the physico-chemical properties of the materials and the respective EC values?

## 2. Materials and Methods

### 2.1. Materials

The biomass precursors (grape pomace and vine pruning) were collected at the University of Hohenheim. They were dried for 8 h at 105 °C. The dried biomass was ground and sorted by a 1.0 mm sieve to provide homogeneity. The model substance microcrystalline cellulose (Sigma-Aldrich, St. Louis, MO, USA) was used for reference purposes.

### 2.2. Methods

#### 2.2.1. Thermochemical conversion

**Hydrothermal Carbonization (HTC).** HTC was conducted in a stainless-steel autoclave (250 mL). The reactor was filled with 25 g of biomass and 125 g deionized water, which is a ratio 1:5. The reaction conditions are listed in Table 1. The autoclaves were placed in a disused gas-chromatographic oven chamber and heated to the desired temperature (heating time was approximately 60 min). The reaction time started when the temperature inside of the autoclave was 1 °C below the desired temperature. After the reaction time was completed, the autoclave was quenched in water (20 °C). The liquid phase was separated by vacuum filtration, using 45 μm filter papers (Whatman). The pH of the liquid phase was analysed using a portable Multi Meter (Hach Lange HQ40D). The solid fraction (hydrochars) was dried overnight at 105 °C.

**Pyrolysis.** Pure biomass samples, cellulose, and previously prepared hydrochars were pyrolyzed in a metal chamber, a simple pyrolysis unit described in a previous study [26]. During the whole experiment, the nitrogen flow was kept at 2 L min^−1^ at 25 °C and ambient pressure. First, the chamber was heated up to 300 °C with a ramp setting of 5 K min^−1^ and maintained for 60 min. Subsequently, the chamber was heated up to 900 °C again with a ramp of 5 K min^−1^ and maintained for 60 min. At the end of the reaction time, the carbonized samples were rapidly cooled down with an increased nitrogen flow of 15 L min^−1^. Finally, the mass loss was determined.

**Nomenclature.** The samples were labeled according to the following principle. HTC denotes that the samples underwent hydrothermal carbonization; the first and second number specify the reaction temperature and time of the HTC, respectively. P900 means that the sample was pyrolyzed at 900 °C. The last part stands for the feedstock materials (Cell = Cellulose; Prun = Vineyard pruning; Pom = Grape pomace). Hereinafter, hydrothermally carbonized biomass will be referred to as hydrochars (HC), pyrolyzed biomass as biochars (BC), and hydrothermally pre-treated biochars as pyrolyzed hydrochars (PHC). 

#### 2.2.2. Analysis

**Thermogravimetric analysis (TGA).** The thermal decomposition of the biomass and of the HCs was performed with a Netzsch STA Jupiter 449 F5. About 20 mg of each sample was weighed into a crucible and placed inside the thermobalance. The samples were heated up to 900 °C with a constant heating rate of 10 °C min^−1^ and a constant nitrogen flow of 70 mL min^−1^. 

**Proximate analysis.** Ash content was determined according to DIN 51719, with combustion occurring in an open crucible at 815 °C. The volatile matter (VM) was determined according to DIN51720, where thermal treatment occurred in a closed crucible at 900 °C for 7 min. The fixed carbon was calculated by difference: 100−Ash−VM.

**Elemental analysis.** The elemental composition of the solid was determined with an Elemental Analyzer (Euro EA-CHNSO) from Hekatech by dynamic, spontaneous combustion and subsequent chromatographic separation.

To compare the results among each other, the ash free content of each element was calculated by Equation (6) with Ex standing for one of the measured elements.
(6)$$E_{X}\left(\mathrm{ash}\;\mathrm{free}\right)=\;\frac{E_{X}\left(\mathrm{with}\;\mathrm{ash}\right)}{1-\mathrm{Ash}}$$

**Carbon yield.** The carbon yield was calculated according to Equation (7); all values are given in percentage.
(7)$$\mathrm{Carbon}\;\mathrm{yield}=\mathrm{Mass}\;\mathrm{yield}\cdot \frac{{\mathrm{Carbon}\;\mathrm{content}}_{\mathrm{Char}}}{{\mathrm{Carbon}\;\mathrm{content}}_{\mathrm{Feedstock}}}$$

**Specific surface area (SSA)**. SSA was assessed using a NOVA e-4200 analyzer (Quantachrome Instruments). To calculate the SSA, multipoint Brunauer–Emmett–Teller (BET) method was applied based on nitrogen adsorption/desorption, carbon dioxide adsorption measurements, and the obtained isotherms [27,28]. Nitrogen isotherms were measured at a temperature of 77 K (liquid nitrogen) and carbon dioxide isotherms were measured at a temperature of 273.15 K. Prior to the measurement, each sample was degased overnight at 100 °C (feedstock and HCs) and 180 °C (BCs, PHCs) under vacuum.

**Electric conductivity (EC) and bulk density (ρ).** The electrical conductivity (EC) of the obtained hydro- and biochars was determined following a procedure described by Celzard et al. [29]. After homogenizing (sieve, 0.1 mm) the samples and drying them overnight at 105 °C, the EC measurement was conducted at room temperature with a device consisting of a glass cylinder and two brass metal plungers (Figure 1). The powder was compressed by the top plunger mass (0.192 kg) and additionally two different weights (2 kg and 5 kg). The resulting pressure (p) was calculated by applying Equation (8): (8)$$p=\frac{F_{g}}{A}=\frac{m_{1}\cdot g}{A}$$
pPressure in Pa.$F_{g}$Weight load in N.$m_{w}$Mass of added weight in kg.gLocal earth acceleration of 9.81 m s^−2^.APiston ground area of 7.9 × 10^−5^ m^2^.

The EC of the compressed powder was studied as a function of the bulk density (ρ), defined by Equation (9), where m was determined by weighting and h was measured with a caliber.
(9)$$\rho =\frac{m_{2}}{A\cdot h}$$
ρBulk density in kg m^−2^.$m_{s}$Mass of sample in kg.APiston ground area of 7.9 × 10^−5^ m^2^.hHeight of sample in the cylinder.

The electrical conductivity σ through the material was of ohmic nature and defined by Equation (10).
(10)$$\sigma =\frac{h}{R\cdot A}$$
σConductivity in S.hHeight of sample in the cylinder in m.APiston ground area of 7.9 × 10^−5^ m^2^.ROhmic resistance in Ω.

The ohmic resistance was measured with a digital multimeter (Voltacraft M-4650B). After determining the blank resistance and height (without and with sample), the weights were applied separately (2 kg and 5 kg). For each weight, the sample mass, height, and resistance were measured. Acetylene Black with a known EC of 400 S m^−1^ [30] was used to calibrate the device and determine its accuracy (±0.5 S m^−1^).

**Compression ratio (CR).** To compare the degree of compression of each carbon material, the CR was calculated by dividing the respective sample volume during the first measurement with P1 (V_P1_) by the sample volume at the third measurement with P3 (V_P3_), applying Equation (11):(11)$$\mathrm{CR}=\frac{V_{P1}}{V_{P3}}$$
CRCompression ratio.$V_{P1}$Volume at applied pressure 1 (blank weight) in m^3^.$V_{P3}$Volume at applied pressure 2 (weight 2) in m^3^.

## 3. Results

### 3.1. Thermochemical Conversion

**Composition of HCs**. In Table 2 the elemental composition, the char-yield, the carbon-yield, and the pH of the process water is given. The elemental composition of the feedstocks is in accordance to known values from the literature [31,32,33,34]. The HTC of cellulose results in a constant HC mass-yield of 45%, whereas biomasses have decreasing yields with increasing reaction severity, here with increasing temperature and reaction time. In addition, the carbon-yield in the solid increases with severity for the cellulose samples, but decreases with the biomasses. On the other hand, the cellulose HC has its highest carbon-content at 70.4 wt.%, whereas pomace and pruning increase until 72.6 and 72.1 wt.%, respectively. 

The resulting pH in the process water is listed in Table 2 and gives information about the production of acids, which are side products of HTC. The highest drop from neutral water to a pH of 1.8 was observed for cellulose. 

In Figure 2 and Figure 3 the van Kreveln diagrams illustrate the difference in H/C and O/C ratios, comparing the precursor compounds cellulose, pomace, and pruning. For pomace and pruning, the ratios are lower than for cellulose. The position of pruning is related to the lignin content, whereas pomace additionally contains tannins [23], which further lowers the ratios. In addition, the graph can represent elemental reactions occurring during the carbonization processes. It is noted that the formation of HCs primarily follows the diagonal lines, according to dehydration [35] (Figure 2 and Figure 3). HCs produced at low operating temperatures (C1, S1; P1; Figure 3) and short reaction times have lost less water compared to biomass, than the HCs formed at higher temperatures and longer reaction time (2–6). Dehydration for cellulose occurs from C1 to C2, following the dashed line. Then, decarboxylation occurs (C2–C6; Figure 3) following the dotted lines. To explain this phenomenon, Hydroxymethylfurfural (HMF) and the resulting HC (hydrochar 1) after dehydration with a theoretical carbon content of 66.7 wt.% are plotted [36]. To increase the carbon content further, decarboxylation reactions are obligatory, resulting in the hypothetical HC (Hydochar 2) after decarboxylation of 1/3 of CO_2_ per HMF molecule (arbitrarily chosen). The step from HMF to Hydrochar 1 represents the major gain in carbon, already achieved by low operating temperature and short reaction time. A gain in higher carbon content, as seen in Hydrochar 2, occurs through slow decarboxylation at higher operating temperatures and extended reaction times. Corresponding results are obtained by Jung et al. [36] on fructose.

The proximate analysis results in Table 3 show an increase of fixed carbon as the carbonization advanced. Consequently, the content of volatile matter decreased. It needs to be highlighted that the ash content in HC is notably lower than in the corresponding biomass. 

**Composition of****PHCs.** Assessing the PHCs by proximate analysis in Table 3, the volatile matter is presumed to be zero, and thus the solid residue is thermally stable and consists only of fixed carbon and ash. Comparing BC from pomace with the respective PHCs, a significantly higher ash content for the BC is noted (15.7 wt.% vs. 3.1–2.4 wt.%); similar observations can be made for pruning, but to a lesser extent (9.6 wt.% vs. 3.4–3.8 wt.%). This can be explained by the stated differences in HC and the precursor and the ash accumulation during pyrolysis [37].

Pyrolysis of cellulose raises the carbon content to approximately 95% with no remarkable differences between BC from raw cellulose and the corresponding PHC. For pomace and pruning the carbon content was also increased by pyrolysis to 93 wt.% and 86 wt.%, respectively. The PHCs from pomace show lower carbon content with 88–89 wt.% due to the higher oxygen content. PHCs from pruning have a higher carbon, and thus a lower oxygen content, compared to the BC from pruning. 

Considering the van Krevelen diagram in Figure 4 regarding the highly carbonized materials, it was observed that the O/C and H/C ratios were highly reduced due to the preferential loss of oxygen and hydrogen, which results in an increase in aromaticity [38]. The present H/C ratios are all ≤ 0.2, indicating high aromaticity in the carbon structure [39]. Furthermore, a direct influence of the HTC on the O/C and H/C ratios of pyrolyzed cellulose is observed. Lower O/C ratios can be achieved with a higher operating temperature during the upstream HTC, while no pre-HTC results in a low H/C ratio. Similarities are found for BC from pruning, since it has a higher cellulosic content. The great shift of PS1 occurs due to ash.

### 3.2. Thermal Decomposition Behavior

The thermal decomposition rate of the precursors and of the different HCs can be followed from the derivative thermogravimetry (DTG) curves (Figure 5). These curves result from the first derivate of the mass loss curve measured with a constant heating rate. The thermal decomposition of the parent materials is depicted in Figure 5A. Pure microcrystalline cellulose shows a single decomposition peak between 300–360 °C, indicating a rapid thermal decomposition [40]. The vine pruning shows the typical decomposition progression of a lignocellulosic biomass—the shoulder between 250–300 °C corresponds to the hemicellulose decomposition, closely followed by cellulose, and the shallower peak between 400–500 °C belongs to lignin. The curve corresponding to grape pomace shows the three peaks characteristic of biomass in addition to a large peak at relatively low temperatures. This peak corresponds to sugars and tannins left after the pressing process [41,42].

Vine pruning underwent a decomposition trend during HTC, which follows a similar path as that of cellulose. In contrast, the grape pomace decomposition occurs considerably faster than the pruning due to the absence of lignin.

### 3.3. Physico-Chemical Properties

#### 3.3.1. Bulk Density (ρ)

Table 4 shows the bulk densities of BCs, HCs, and PHCs obtained at 646 kPa. The bulk density of HCs obtained from cellulose decreases with higher operating temperatures and extended reaction times. The density of the pruning does not change significantly during HTC and ranges between 0.48–0.50 kg m^−3^. Results of *ρ* for BC from cellulose and pruning and the effect on *EC* are discussed in Section 4.3.

#### 3.3.2. Specific Surface Area (SSA) 

Comparing the SSAs of the different materials (Table 5), a huge difference between precursor materials (BM), HCs, BCs, and PHCs is observed. The HCs do not show remarkable differences between each other and the SSAs are in good agreement with the content of volatiles (the lower the volatiles content, the higher the specific SSA and H/C molar ratio). As shown in Table 5, SSA decreases with the reaction time. This decrease in SSA becomes more significant at higher temperatures, as shown in Figure 6.

**Hydrochars (HCs) from cellulose and pruning.** The nitrogen adsorption isotherm obtained for HCs showed multi-layering behaviour according to the IUPAC (International Union of Pure and Applied Chemistry) classification type II [44], with a concave tendency at low pressures and at increasing pressures the isotherm becomes first linear and finally convex. This isotherm form is observed because at lowest pressures, the nitrogen molecules are adsorbed into the few micropores, then diffuse into the mesopores and macropores with increasing relative pressure (first monolayer at moderate pressure, then condensed multilayers under high pressure). The physisorption isotherms presented in Figure 7A,B can be considered as representative for all other HC samples from cellulose and pruning, respectively.

**Biochars (BCs) and pyrolyzed hydrochars (PHCs) from cellulose.** The BC from raw cellulose also showed a significantly lower SSA compared to the corresponding PHCs (Table 5). It needs to be highlighted that the nitrogen physisorption isotherm (Figure 8A) shows stepwise multi-layer adsorption on BC from raw cellulose, assigned to an isotherm of type VI. Pyrolysis led, as expected, to an increase in SSA for all samples. The latter is confirmed by the results of BET measurements with CO_2_ as an adsorbate gas, since higher SSA values were measured (see Table 5). This aspect will be further discussed in Section 4.3 as well as the SSA of the PHCs and their influence on compressibility and EC. 

For PHCs from cellulose, the nitrogen adsorption/desorption isotherm can be assigned to type IV, as shown in Figure 8B. 

**Biochars (BCs) and pyrolyzed hydrochars (PHCs) from pruning.** The nitrogen physisorption isotherms for P900-Prun and HTC-240-60-P900-Prun are presented in Figure 9A,B. The isotherm of BC from pruning shows Type III characteristics. 

As shown in Table 5, the SSA of BC from pruning at 25 m^2^ g^−1^ is surprisingly low. A pre-treatment with HTC can remarkably increase the surface.

The absolute values of the SSA calculated based on the physisorption isotherms for CO_2_ are included in Table 5. It can be observed that for BCs and HCs, as well as for PHC from pruning, significantly higher SSA values are obtained via CO_2_ adsorption, whereas for PHC from cellulose, this is not the case. This correlates well with the measured pore size distribution (PSD) (Figure 10), which shows a high content of micropores (pore diameter < 2 nm) for all carbons. Meso- and macropores are observed only for HCs (Figure 10A), and to a much smaller degree also for BC and PHC from pruning (Figure 10C), whereas PHC and BC from cellulose show only microporous characteristics (Figure 10B).

### 3.4. Electric Conductivity (EC) and Physisco-Chemical Properties

**Cellulose.** The EC results obtained for BC and PHC from cellulose are shown in Figure 11A. All BCs and PHCs show semi-conductive properties, since EC is less than 10^4^ S m^−1^ but higher than 10^−8^ S m^−1^ [45]. Furthermore, the EC of the biochars behaves proportionally to the applied pressure, independently of the pre-treatment of each biochar. Comparing the EC values of all individual materials with each other, it needs to be highlighted that all PHCs show higher ECs compared to BCs obtained via direct pyrolysis of cellulose. 

The diagram presented in Figure 11C is a synthesis for both previous diagrams, illustrating the correlation between EC and ρ obtained at a p of 645 kPa for different temperatures and reaction times. In general, higher HTC temperatures lead to lower EC, lower ρ values, and higher CRs (Figure 11D).

**Pruning.** The EC results obtained for BC from raw pruning and the corresponding HCs are shown in Figure 12A. All samples show semi-conductive behaviour. In addition, the proportional rise in EC is observed again. However, slight differences concerning EC of biochar (P900-Prun) to pre-treated biochar (HTC-x-x-P900-Prun) can be observed. 

The higher EC of pre-treated biochar in comparison to pyrolyzed pruning can be explained by the leaching of inorganic materials during HTC, as stated in Section 3.1. As referred by Barroso-Bogeat [46], a higher ash content in the carbon material leads to a low EC, which is why biochar from raw pruning shows the poorest EC in Figure 12A. 

The results for the ρ of BC from pruning and PHCs versus applied p are shown in Figure 12B. For all BC samples, the ρ is increased by applying a higher p. It is remarkable that by increasing the applied pressure, the differences in ρ decrease, whereas the differences in EC increase among all samples (Figure 12A,B). This effect could not be observed for the BC and PHCs from cellulose (Figure 11A,B) due to the lack of lignin.

Figure 12C,D shows the comparison of EC and ρ, as well as the CRs of the BC and PHCs, respectively. This implies that the compression does not affect the EC of the BCs among themselves, but rather the intrinsic EC of the carbon particles seems to be decisive. In contrast to cellulosic PHCs, no correlation between EC and ρ or pre-HTC conditions are found for PHCs from pruning. Nevertheless, it needs to be highlighted that P900-Prun shows characteristics of pyrolyzed cellulosic hydrochar in terms of compressibility (CR). Due to the high content of cellulose in pruning (see Section 3.1 and Section 3.2), the biomass, to a certain degree, behaves according to the pattern described for BCs from cellulose. However, the lignin fraction follows a different conversion pattern, which cannot easily be compared to that of cellulose, and is further discussed in the following considering the results of the adsorption experiments.

## 4. Discussion

### 4.1. Thermochemical Conversion

The solid-mass loss during HTC can firstly be attributed to the solubilization of the feedstock after hydrolysis, furthermore dehydration, aldol-condensation that releases H_2_O, as well as decarboxylation that releases CO_2_. Aldol-condensation results in hydrochar formation and increases carbon-yield as well as the carbon content [36,47], which is in contrast to decarboxylation, where an increase in carbon-content is accompanied with a loss of carbon into the gas phase. With increasing reaction severity a shift towards decarboxylation can be observed in the van Kreveln diagram (Figure 4), and consequently the char- and carbon-yield is expected to drop. As can be seen in Table 2, this is not the case for cellulose—char yield remains constant and carbon-yield increases. The expected loss is, therefore, compensated by reactions that increase mass-and carbon-yield. 

In a previous contribution about the HTC of fructose, it was demonstrated that the carbon yield after 250 °C reaction temperature is 62%, whereas 200 °C only delivered 50% [36]. Based on that observation, it is assumed that the yield of aldol-condensation of HMF, which forms hydrochar, increases with reaction severity, and therefore compensates carbon losses through decarboxylation to a certain amount. Lu et al. also found similar results with the HTC of cellulose, however they also showed that higher temperatures (e.g., 250 and 275 °C) resulted in remarkable carbon losses in the solid in the long term (reaction time > 20 h) [48], as well as with a reaction temperature of 280 °C [49]. 

During the conversion of the biomass to HC dehydration and decarboxylation reactions take place simultaneously, visualized from P1 to P6 and S1 to S6 in Figure 3. For the conversion of pomace the carbon losses and the trends in the van Kreveln diagram indicate that decarboxylation strongly takes place. It can be assumed that this is due to a lower content of convertible carbohydrates. In addition, the protective effect of the lignocellulose structure hinders the degradability of the carbohydrates (supported by TGA results in Section 3.2). Considering the results of the proximate analysis in Section 3.1, especially the pomace loses a significant amount during its conversion, from 5% to approximately 2%. Ash decreases in pomace HCs more than in pruning, due to the feedstock-specific ash chemistry. This is also observed in several studies [50,51,52] and is assigned to leaching into the process water at high temperatures and in an acidic environment [53].

The efficiency of the HTC of cellulose can be increased with reaction severity, giving the opportunity to conserve more carbon into the solid. This is also beneficial for the further pyrolysis. In each case, HTC of cellulose had a positive impact on the carbon-balance compared to simple pyrolysis without HTC. Interestingly, this is the opposite for the biomasses—char-and carbon-yield decrease with increasing reaction severity, and C-efficiency after pyrolysis is lower compared to pyrolysis without HTC. Especially, pomace strikes out with a large carbon loss when HTC is combined with pyrolysis. The decarboxylation during HTC, therefore, results in a negative impact on the overall carbon-balance. For pruning the trend is similar, however, not as strong—the carbohydrates compensate the carbon losses that accompany with decarboxylation. Interestingly, the carbon-content of the biomass HC (72.6 and 72.1 wt.%) is higher than for cellulose HC (70.2 wt.%) at the highest reaction severity. This is probably coming from the decarboxylation, which increases the carbon-content right from the beginning for the non-hydrolysable compounds of the biomasses. Cellulose, in contrast, needs to undergo hydrolysis, dehydration, and char-formation before it can start with decarboxylation. The trend in the van Kreveln diagram strongly supports that conclusion, as cellulose HC firstly moves along the dehydration line and later moves over to decarboxylation.

The nitrogen content increases with a higher operating temperature and extended reaction time, inferring that most of the nitrogen is retained in the solid, and is concentrated as the amount of HC decreases [54]. However, the HTC of pomace and pruning represents an effective treatment to reduce the ash-content, which can be a major advantage for the use in a DCFC. 

The drop of the pH observed for the process water of cellulose (Section 3.1, Table 2) suggests formation of levulinic-, formic-, and acetic-acid [55]. This drop also occurs but is much less distinctive for pruning and pomace, suggesting that leached inorganic basic compounds buffer the decrease in pH [56]. It can also be assumed that the carbohydrates are a major source of acids, making process water from cellulose HTC more acidic than the biomasses, thereby pomace, with the lowest amount of carbohydrates, results in process water with the highest pH value. Furthermore, an increase in pH with a higher operating temperature and extended reaction time is observed. This can be explained by the decomposition of acids [57,58,59]. 

Regarding the composition of the PHCs, it can be stated that in general, the oxygen content is reduced during the pyrolysis due to decarboxylation, but as shown in Table 2 (Section 3.1), the PHCs from pomace have an increased oxygen content compared to the BC without pre-treatment. This could indicate a richer surface chemistry, depending on the present oxygenated functional groups [60]. However, HTC on pruning reduces the oxygen content in the resulting BC. The nitrogen content in the BCs is rising and can be attributed to the formation of heterocyclic compounds with a bound N, such as pyridines and pyrroles [61].

### 4.2. Thermal Decomposition

Regarding the thermal degradation behaviour, the DTG curve (Figure 5 in Section 3.2) serves as a valuable tool to understand the decomposition process that the biomass underwent during HTC. Cellulose, for example, undergoes a series of structural changes as a result of the hydrolysis and repolymerization reactions during HTC. From the DTG, it is concluded that a temperature of 220 °C and reaction times considerably longer than 2 h are necessary for a full conversion of cellulose. On the other hand, it appears that the effect of the reaction temperature used during the HTC has a stronger effect on the degradation of the cellulose than reaction time. The DTG of the HC from cellulose obtained at 220 °C after 120 min showed three peaks, with maxima at 250 °C, 330 °C, and 410 °C. At higher temperatures and longer reaction times, the second peak disappears, indicating a complete conversion of the cellulose into HC. This agrees with the results obtained by Simsir et al. [62]. The peak at 250 °C is possibly a result of the decomposition of intermediate compounds formed during the cellulose hydrolysis (cellobiose, cellotriose, etc.) [63,64] or the HMF polycondensation. Interestingly, this peak slightly decreases with the increasing severity of the parameters, which is an indication that this is an intermediate compound, which is quickly formed but slowly decomposes into HC during HTC.

Pruning has a high cellulose content, being a wood-like biomass [65]. However, the presence of lignin has a shielding influence (mentioned in Section 3.1), which leads to the slower decomposition rate of the polysaccharides. This is evidenced by the peaks shown by the HCs obtained at 220 °C and 300 min, as well as 240 °C and 60 min. With these parameters, the crystalline cellulose was completely converted, whereas the pruning still showed the cellulose peak. The right shoulder present at temperatures around 400 °C corresponds to the HC, as well as to lignin. The broad peak is a consequence of the different bonds present between the three building blocks of lignin [66] and the structure complexity of the HC structure composed of aromatic and semi-aromatic rings [67,68].

As already stated in Section 3.2, the grape pomace has a very low lignin content, and the content that is available comes from the seeds. Furthermore, it has a high concentration of simple sugars and cellulose. These facts explain the faster decomposition of grape pomace during HTC. 

### 4.3. EC and Physico-Chemical Properties

In this paragraph, the EC and physico-chemical properties, including indicators such as SSA, ρ, or CR, are discussed to characterize the obtained materials with regard to their possible application. 

In this context, it is important to keep in mind that the electrical properties depend on microscopic parameters (the molecular structure, bonding between the carbon atoms, intrinsic EC of the particles) and macroscopic parameters (applied pressure p, bulk density ρ, compression ratio CR). Since biomass usually contains only σ-bonds in the sp^3^ state [69], no EC was detected for the precursor materials. The corresponding HCs do not show any EC as well. The reason is that HC indeed has sp^2^ carbon atoms and double bonds, but the pseudo-aromatic furfural rings π orbitals are not connected, therefore, no large π-systems with delocalized electrons exist. An electric conductivity along the surface is also hindered by the high amount of oxygen and nitrogen (Section 3.1), which implies many functional surface groups counteracting the EC. Hence, the biomass and HC samples are insulators with an EC less than 10^−8^ S m^−1^ [45]. Only the samples obtained via pyrolysis (BCs and PHCs) possess the desired EC, confirming that pyrolysis leads to lower electrical resistance (also observed by Radeke et al. [70]), because of increased aromatization. In addition, the volatilization of surface functional groups and higher SSA values, correlating with higher bulk densities, support electron transport via the surface. 

The density of a sample is an intrinsic property and must be regarded as the bulk density ρ of an isotropic packing, since voids between the carbon particles are still present. In theory, applying pressure on such a packing will deform the macroscopic disordered system, which is schematized in Figure 13. As shown in Equation (10) in Section 2.2.1., EC is the reciprocal value of the electrical resistance R. The overall electrical resistance of a carbonaceous powder, in turn, depends on two different factors: the intrinsic electric resistance of a single carbon particle and the total size of the contact area between the particles [71]. Thus, the EC is strongly influenced by the packing of the particles. A higher p forces the carbon particles closer to each other, resulting in more surface contacts (=higher contact area), and hence an improved EC. This has already been observed by several researchers [21,29,71,72,73,74].

It can be assumed that the applied pyrolysis led to a hybridization of the carbon atoms resulting in π-bonds in the sp^2^ state [69], which implies more delocalized electrons available as charge carriers. The development occurs only partially and not every sp^3^ is converted into a sp^2^ carbon atom, forming π-bonds. Consequently, the carbon materials are transformed to a partly graphitized form [75]. This is confirmed by the low H/C ratio stated in Section 3.1. Additionally, the oxygen content, indicating inter alia the amount of oxygen-containing surface groups, decreased remarkably, which favours the increase of EC due to the elimination of insulating effects caused by the functional groups on the surface [76].

This decreased oxygen content combined with the polymerisation during HTC explains why the PHCs show higher EC values then BCs for cellulose and real biomass. Regarding pruning, this can be explained by the already existing poly-aromatic cross-linked lignin structure. As stated in Section 3.2, pruning undergoes only a partial conversion during the hydrothermal pre-treatment due to the lignin content. For cellulose, the effect can be explained by the microcrystalline and partly amorphous structure of cellulose, which is why the pyrolysis of cellulose will lead to the conversion of aliphatic carbon into an aromatic carbon structure [77]. In contrast, the polymer obtained from HTC has already formed aromatic clusters [78] with interconnections [43]. Applying pyrolysis on such a complex structure will then lead to a higher organisation due to aromatization. A structure with a higher aromatization level has more delocalized electrons available due to a higher amount of the above-mentioned sp^2^-hybridized carbon structures [69]. 

Thus, the pre-treatment with HTC of PHCs leads to the formation of clusters with π-bondings in the carbon structure (see Figure 14), which finally leads to a higher EC of the pre-treated biochars (PHCs) compared to the biochars obtained via direct pyrolysis of biomass (BCs).

Comparing the EC of the PHCs from Cellulose with each other (Figure 11A), HTC-220-120-P900-Cel has the highest EC, approximately three times higher than HTC-260-60-P900-Cel. However, the temperature of the HTC treatment seems to be the determining factor of the final EC value of a biochar, since the higher the HTC temperature, the lower the EC value of the final biochar, independent of the reaction time. 

This could be explained by the decrease in ρ with higher HTC temperature, since the lower the ρ, the lower the particle contact area, and thus the EC value [21,79]. However, this assumption is contradicted by the observation that the influence of temperature on EC seems to be much higher than the influence of the ρ on EC (see Figure 11C). Considering this and the suggested production pathways in Figure 14, it can be assumed that the measured EC value is mainly determined by the intrinsic EC of the particles, and to a minor extent by the particle contact area between them. The results for the ρ of pyrolyzed cellulose (BC) and PHCs from cellulose versus the applied pressure p in Figure 11B confirm the expectation that the ρ is determined by the applied p. However, this is not the case for the BC from cellulose P900-Cel, since the ρ does not significantly change with p. In addition, the CR value of P900-Cel at around 1.08 represents the lowest CR value among all samples, suggesting a very low compressibility (Figure 11D). In combination with the low EC values, these results suggest that structural properties of the BC particles or a low intrinsic EC of those particles are the determining factors for the low EC, and not the physical indicators, such as CR or ρ. 

For the PHCs, the EC seems to be influenced mainly by the ρ and the intrinsic EC, not by compressibility expressed by the CR values (because of no clear trend in Figure 11D). Especially, those PHCs pre-treated with higher HTC temperature showed lower EC values due to previous decarboxylation reactions (compared to the BCs), as shown in the van Krevelen diagram in Section 3.1. In theory, the decarboxylation breaks down the HC polymer, resulting in a molecular structure with more defects, and hence lower EC values.

In addition, it needs to be highlighted that the previously observed decrease in ρ with higher HTC temperature and extended reaction times in the HCs is also reflected in the BCs (Figure 11B). The difference in ρ of HC to the corresponding BC is a result of volatilization and associated increase in porosity (confirmed by SSA values in Table 5, Section 3.3.2) [80]. 

The fact that lower bulk densities lead to lower EC values corresponds well with the literature, since Sánchez-Gonzáles et al. [73] and Hoffmann et al. [21] made similar observations and refer to the likelihood of contact between the carbon particles under higher compression. However, it is worth noting that the impact of the reaction time on the ρ and the EC value changes with the temperature. While at 220 °C, ρ and EC values decrease significantly with a higher reaction time, this is not observed at 240 °C, and at 260 °C the EC even increases with higher reaction time. Whereas the commutated development of EC and ρ at 220 °C correlates well with the literature (see above), the change of the EC values at 240 °C and 260 °C with the reaction time cannot be explained by the change in ρ, since the ρ stays the same. 

It is assumed that the change in SSA is the main reason for the dependence of EC on reaction time and that the coalescence of carbon spheres accelerated by higher temperatures and reaction times is the reason for the decrease in SSA, since this effect has already been observed by Jung et al. for HCs from fructose solutions [36]. According to Sevilla et al. [78], HCs usually consist of amorphous structured macrospheres, and hence show a relatively low SSA.

The SSAs obtained for BCs and PHCs from cellulose correspond well with literature, since similar SSAs were obtained by Kang et al. [81] on pyrolyzed cellulosic hydrochar. The type IV isotherms observed for PHCs from cellulose (Figure 7B) indicate the presence of macropores. Due to the open hysteresis loop of desorption isotherms, capillary condensation on constrictions in the porous network, probably associated with disordered carbon, and the presence of inkbottle or narrow slit-like pores (see Figure 15) are assumed. 

The stepwise multi-layer adsorption on BC from cellulose (isotherm type VI, see Figure 7A) lead to the assumption that P900-Cell has a uniform non-porous surface (as observed for graphitized carbon black) [44], but due to the open hysteresis loop, the possibility of ink-bottle pores have to be considered, where the nitrogen molecules get trapped after they enter the pores and subsequently cannot be desorbed again [82]. Other possible reasons for this open hysteresis loop could be the adsorption potential of the carbon towards nitrogen molecules at such a low temperature or a deformation of the carbon structure due to the adsorption process (e.g., formation of closed pores) [82]. However, the high SSA values obtained from adsorption measurements with CO_2_ lead to the suspicion that a high content of micropores or ink-bottle pores are present in the carbon structure. This assumption is further supported by the PSD in Figure 10B, showing the high content of micropores.

The type III isotherm observed for BCs and PHCs from pruning are uncommon according to Sing et al. [44,75] and an indication for strong multilayering and no formation of a complete saturated monolayer. This assumption is also supported by Rouquerol et al. [83], where Type III isotherms are related to relatively weak adsorbent-adsorbate and strong adsorbate-adsorbate interactions, which leads to the convex form of the isotherm with the quantity of gas adsorbed < 2, even at higher p/p° values. In general, this is observed for nonporous or macroporous solids, where the monolayer coverage is overtaken by a form of co-operative multilayer adsorption, in which molecules are clustered around the most favourable sites [83].

This is why mainly macroporosity is assumed, supporting the previously discussed theory of blocked microstructures. The isotherm of the PHC, on the other hand, shows Type IV characteristics (Figure 8B). Again, the open hysteresis loop of the desorption isotherm and the PSD in Figure 10B indicates a microporous structure, further confirmed by the slight Type I characteristics at lower pressures. However, the wide shift of the last adsorption pressure is again an indication for macropores and mesopores [83], which is again confirmed by the PSD results in Figure 10B.

The small SSA and the observed multi-layering behaviour in the case of nitrogen adsorption (type II isotherms according to IUPAC classification) in combination with the higher SSAs obtained based on CO_2_ adsorption isotherms allow the conclusion that the obtained HCs already contain meso- and microporosity to a certain extent. This is also confirmed by the PSD in Figure 10A. Dieguez-Alonso [84] et al. state that mesoporosity is independent of the raw materials, the operating temperature, and reaction time. However, the lack of a hysteresis loop at high relative pressures is contradictive, since such a hysteresis loop would be a typical indication for mesoporosity in carbons [85]. This observation and the obtained PSD shown in Figure 10A lead to the assumption that the HCs mainly have a microporous structure with some meso- and macropores. The increase in SSA after pyrolysis is assumed to occur due to the removal of condensed species and volatiles [86] and the formation of micropores.

Furthermore, it is assumed that inorganic species provoke partial blocking or constrictions of the porous structure [87,88], as well as the presence of ink-bottle and slit-like pores (Figure 15) in the HCs and the BCs (P900-x). The assumption of pore blocking due to secondary reactions between the tar produced during pyrolysis and the solid at high temperatures, which was assumed by Dieguez-Alonso et al. [84], seems to have no major significance here.

Since adsorption of nitrogen molecules under the measurement conditions is hindered [82], adsorption measurements with CO_2_ are recommended [44,83] and were conducted to get more information about the pore structure.

Although a direct comparison between SSA values obtained from N_2_ and CO_2_ adsorption is not possible, as the measurement conditions are highly different (temperature and its influence on the molecular kinetic energy of the gases [82]), the combined analysis of the different SSA values allows more precise statements about the respective materials. Thus, considering Figure 16, it can be shown that the above-mentioned assumption concerning ink-bottle and slit-like narrow micropores in BCs and PHC from cellulose can be verified based on the obtained SSA data (CO_2_ adsorption much higher than N_2_ adsorption) and PSD results (Figure 10), whereas for the PHC from pruning (HTC-240-60-P900-Prun) and the HCs, a high content of micropores in a meso- or macroporous structure is more likely. Thus, it can be concluded that the pore formation during different processes highly depends on the precursor biomass; for cellulose, one-step processes (pyrolysis or HTC) and their combination lead to the formation of microporous structures, for pruning the combined process leads to the formation of smaller micropores at the expense of meso- and macropores, while pyrolysis leads to slightly bigger micropores and a higher content of meso- and macropores. This pore structure in PHC from pruning is further confirmed by the higher SSA value from N_2_ adsorption and the lower SSA value from CO_2_ adsorption compared to that of the corresponding BC from pruning (Table 5).

The above-mentioned assumption concerning BCs and HCs is supported by Sing et al. [44] and Evans [90], where the hysteresis behaviour on narrow slit-like pores is extensively discussed. The particles have internal voids with an irregular shape and broad size distribution. In this case, the type I isotherm character of adsorption at lower pressures is indicative of microporosity, which is additionally confirmed by the presence of the open hysteresis loop at low pressure and the PSD results. Residues of N_2_ in these micropores can only be eliminated by outgassing at an elevated temperature [82]. Sevillia et al. [91] were able to eliminate the hysteresis by removing the amorphous carbon in the structure of graphitized carbon from cellulosic HC, supporting again the previous developed hypothesis. 

### 4.4. Application in Energy Storage and Conversion Devices (EDLC and DCFC)

After presenting the physicochemical and electrical properties of the obtained carbon materials from vineyard residues, this paragraph provides a critical assessment of the materials regarding their possible application as a fuel in DCFCs or as electrode materials in EDLCs. 

As previously mentioned, carbonaceous materials have been tested as fuel in DCFCs in order to reveal their efficiency as energy carriers or as bio-based electrode materials in EDLCs as a sustainable substitute for activated carbon from fossil sources. Nevertheless, the required material properties for each of these applications differ considerably due to the different underlying electrochemical mechanisms (electrochemical energy conversion vs. electrochemical energy storage) and the fact that in DCFCs the carbon materials serve as a fuel and are used energetically, whereas in EDLCs the materials are used materially. By discussing the properties of the obtained chars with regard to these two promising but different applications in the field of energy conversion and storage, new insights into biomass potentials are given by showing new possible applications in future-oriented technologies.

Regarding DCFCs, it has been found that the physicochemical properties affect notably the electrochemical reactivity of carbon materials within the active electrochemical zone (AEZ) and subsequently the DCFC lifetime. Several researchers proved that the overall carbon fuel cell efficiency is ascribed mainly to gas-AEZ interactions rather than to the extremely pure carbon-AEZ contact. The reverse Boudouard reaction (C + CO_2_ → 2CO), which is the main non-electrochemical reaction probable to take place at the high operating temperature range of DCFCs (>700 °C), and thus has demonstrated a key role in the DCFC performance as its gaseous product (CO), can easily diffuse within the AEZ much more rapidly than solid carbon, contributing to power generation. Kaklidis et al. [92] clearly correlated the CO formation rate in open circuit conditions with the achieved DCFC electrochemical performance and proved that the overall chemical and electrochemical processes are driven by the CO shuttle mechanism. 

Regarding EDLCs, as well as DCFCs, some physicochemical properties showed a direct relationship with the delivered performance, such as the volatile matter, the carbon-hydrogen-oxygen (CHO) content, and the carbonaceous materials structural disorder. Whereas for EDLCs, it has been shown that a low content of volatiles, high content of carbon (>90 wt.%), the presence of heteroatoms, such as oxygen or nitrogen, and an ordered, graphitized carbon structure is favourable for the capacitive performance [17,93], in the case of DCFCs, the carbon content is a key property that gives insight into the utilization of the carbonaceous materials as fuel in DCFC, and hence the performance of the system. It appeared that carbon content higher than 90 wt.% is not preferred in DCFC systems as it hinders the generation of light gases, such as CO and H_2_. The carbon content should be within 70 wt.% and 85 wt.% range in order to achieve good electrochemical performance [9,94,95]. As a consequence, the obtained HCs with their lower C content seem to be more suitable for use in DCFCs, whereas for the application in EDLCs, the PHCs with C contents higher than 90 wt.% are more appropriate.

Concerning the oxygen content, the presence of oxygen functional groups could be suggested and even evaluated, which has a positive impact on the electrochemical reactions within the DCFC [96]. The various surface species referenced in Section 3 have different influences on the equilibrium potential of the fuel cell. Their diverse transient desorption behaviours from the carbon surface might be the reason [97]. The higher the oxygen content within the carbonaceous materials the more CO is formed and the better the power generation in the fuel cell. The oxygen content should be higher after the pyrolysis of vineyard pruning and pomace biomass and HCs compared to values presented in Table 2 and is preferred to the approach at 25 wt.%. Conversely, for EDLCs lower oxygen contents are preferred, since high content of heteroatoms lead to decreasing EC values, which negatively influences the performance [98]. Again, according to the values given in Table 2, HCs with their high oxygen content would be suitable as fuels in DCFCs, whereas the lower oxygen content of BCs and PHCs (<10 wt.%) would be favourable in an EDLC, since similar contents have been shown to give rise to pseudocapacitive effects, which increase the total capacity of the EDLC [11].

In the light of the elemental composition presented in Table 2, it seems that decreasing the pyrolysis temperature to 500 °C would lead to biochars with higher oxygen contents and slightly lower carbon contents that could fit the DCFC operation [9,94]. Additionally, the Higher Heating value (HHV) can be estimated by taking the hydrogen and oxygen content into account (apart from the carbon content) as a basis. Based on the above mentioned discussion of the desired elemental composition of the vineyard residue biochar for its use as fuel in a DCFC, it appears that the HHV will be lower than that of all the PHCs from vineyard pomace and pruning prepared at 900 °C, but it will not present a significant effect on the DCFC performance.

In terms of carbon yield, higher operating temperatures and extended reaction times during the HTC pre-treatment may have a deleterious effect on biomass precursors, since carbon, and hence potential energy, is lost. This tendency in terms of carbon yield is more pronounced when the pyrolysis is carried out at 900 °C, which proved again the need for lowering the pyrolysis temperature to fit the DCFC operation requirements. For the application in EDLCs, in contrast, the high pyrolysis temperature of 900 °C seems to be the decisive factor with respect to EC and SSA, as shown in Table 4 and Table 5. Since high surface areas of the electrode materials are one of the key properties for electrostatic energy storage, and the graphitization effect of the carbon materials increases with higher temperatures [21], a lower pyrolysis temperature is not recommended when aiming at EDLC applications.

The absence of sulphur in all samples is beneficial, since it poisons the anode materials, such as nickel or copper, especially in a Ni-based catalyst DCFC. Nitrogen content in the samples is relatively low, but in DCFCs N-doped graphene layers are assumed to contribute to a higher chemical reactivity toward oxygen due to more edge-active sites [99], and N-doped electrode materials lead to higher capacities in EDLCs, since N acts as an electron donor, which leads to higher pseudocapacities [19,100]. Although, according to literature, higher N contents would be favourable, the relatively low N content is a consequence of the selected biomass, and could be increased by using N-rich biomasses as a precursor or adding other N-sources, such as urea, during the carbonization or activation processes [101].

In terms of volatile matter (VM), no consensus has been found yet regarding DCFCs. Chien et al. [102] reported that a high VM content causes a reduction in performance, while a medium VM content is best suited for a fuel, but does not classify the quantity medium further. Arenillas et al. [92], contrarily, mention that volatiles greatly enhance the DCFC performance, probably because of an improved reactivity of coal and disordered structure. Jang et al. [8] claim that volatiles consist of electrochemically oxidizing products, such as H_2_, CO, and hydrocarbons, which may enhance the performance by lowering the cell resistance, as discussed previously. Performance also increases, since gaseous products easily access the three phase boundary (TPB) reaction sites, expanding the contact area [103], and enhance the electron transport [104]. 

In this study, the pyrolyis was adopted to prepare the biochar issued from pomace and pruning vineyard biomass and HCs. It is well known that the pyrolysis process leads to a decrease in the VM content of the biochar compared with the raw biomass. However, the reported VM contents were nil after carrying out pyrolysis experiments for both vineyard pruning and pomace biomass and HCs (Table 3). This could be related to the high pyrolysis temperature (900 °C) leading to a full devolatilization of the biochars favouring mainly fixed carbons (FC) at the expense of oxygen functional groups. Konsolakis et al. [105] showed that the biochar reactivity within the anode of DCFC is hindered by an increased fixed carbon. It should be preferred to perform pyrolysis of the vineyard residues, but at lower pyrolysis temperatures that do not exceed 500 °C. It has been previously proven that promising performance of a DCFC system fed directly by olive wood and almond shell biochars, having higher VM contents, is possible [9,94]. 

When emitting volatile matter from the inner area of raw biochar particles, many small-sized pores are generated, which change the surface properties; this effect has been discussed in the previous section. The devolatilization and low VM content is favourable for EDLC applications, since it is accompanied by a higher carbon content and higher EC values [98].

Considering the ash content, the literature is unanimous. A low ash content is desirable, especially for DCFC based on solid oxide electrolytes with specialised and precise anode materials. The inorganic’s slagging propensity reduces the lifetime by blocking the active sites of the anode [106]. Several authors analyzed the mineral composition of ashes and showed that some minerals may act as catalyst (K_2_O, CaO, Fe_2_O_3_), whereas other works as inhibitors (SiO_2_, Al_2_O_3_) for the chemical and electrochemical reactions taking place within the anode of DCFC [9,94]. For example, the presence of CaO within the ash of a biochar leads to catalytic effects. This oxide acts on the equilibrium of Water Gasification Reaction (WGR), which is accelerated in the forward direction to produce more CO and H_2_ [9]. Furthermore, CaO demonstrated an effective role on the CO_2_ sorbent following the carbonation reaction (CaO + CO_2_ → CaCO_3_), and thus is considered as a heat carrier that provides the necessary energy for the endothermic gasification [9]. Meanwhile, SiO_2_ can alternatively impede the carbon anodic reaction inside the DCFC due to the possible formation of isolating film of SiO_2_ on the anode reaction surface. 

Similar observations have been made concerning metal oxides in electrode materials for EDLCs, since some metal oxides are characterized by remarkable high pseudocapacitive properties (e.g., RuO_2_, Fe_3_O_4_ or MnO_2_) [107,108,109,110,111]. Nevertheless, the assessed biomass precursors do not contain these desireable metal-oxides and other inorganic compounds should be as low as possible, since they reduce the accessible SSA by pore clogging.

The electrochemical reactions in a DCFC occur predominantly on the carbon surface represented by the pores. A higher porosity consequently results in a larger SSA. As stated before, a high interfacial SSA is more available for electrochemical reactions and can enhance the efficiency of the DCFC. This provides a major advantage for HCs and resulting biochars over other fuels with smooth surfaces (e.g., raw biomass, graphite). Especially, the suspected higher vulnerability to pore clogging by inorganics and tars makes un-treated biomass a poor carbon fuel. With regard to the introduced fuel cell technologies, the DCFC-based molten carbonate electrolyte and the hybrid DCFC benefits more from the porosity of the carbonaceous skeleton of the charcoal fuel. The liquid carbonate is sometimes mixed with carbon materials within the anode of the hybrid DCFC and can diffuse even better into the pores. In contrast, the DCFC, based on solid electrolyte efficiency, suffers to an increased carbon loss by the Boudouard reaction with high surface carbon fuel [112].

According to Table 5, the SSA obtained via CO2 adsorption of vineyard pruning and cellulose biochars prepared without passing the HTC process is 414 and 711 m^2^ g^−1^, respectively. These SSA values are promising compared to the pyrolyzed HCs from both samples, showing that biochar is preferred over pyrolytic HC due to better availability of reaction sites within the anodes of DCFC. For the application in EDLCs, the obtained SSAs from CO_2_ are very promising, since they show the presence of microporosity within the biochars, which has been proven to be an advantage for application in EDLCs. Nevertheless, the SSA obtained via N_2_ adsorption for the biochars is relatively low, with only 111 m^2^ g^−1^ (P900-Cell) and 25 m^2^ g^−1^ for biochar from pruning (P900-Prun), which leads to the assumption that the obtained SSA for pyrolytic biomass is too low for an application in energy storage devices, which are based on the formation of a double-layer on the carbon surface. On the other hand, the SSAs obtained for pyrolitic HCs (PHCs) from cellulose are very promising, since they range from 319 to 441 m^2^ g^−1^ for N_2_ adsorption, and additionally possess microporous structures proved via CO_2_ adsorption measurements. Unfortunately, these SSAs are not reached when HCs from real biomass, such as pruning, are pyrolyzed. However, it is very interesting that despite the low SSA obtained via N_2_ adsorption for PHC from pruning (43 m^2^ g^−1^), the SSA obtained via CO_2_ adsorption is as high as the SSA obtained for PHC from cellulose. This effect hast been discussed in Section 4.3 extensively and its influence on the suitability as an electrode material needs further investigation. Testing the materials in an EDLC to compare their electrical performance is crucial in this context. Until now, it has been assumed that the PHCs from cellulose and pruning would lead to good EDLC performances due to their microporous structures.

The EC of the carbon fuel plays a role only in the case of its use in a pure individual MCFC, since the carbon fuel acts as an anode. A high EC can lower the ohmic polarization of the anode [113] and increase the electron transport to the current collector. As mentioned, the DCFC based on SOFC and the hybrid of SOFC/MCFC electrolytes have a metal current collector acting as an anode, thus both technologies are independent of EC. However, the EC permits one to make assumptions about the carbon molecular structure, disorder, and the content of oxygen functional groups. The lower the EC, the higher the content of disordered structures due to the improved voids between carbon particles, and subsequently the decreased particle contact and pathway for electron transfer. This may be correlated to the contact theory given by Mrozowski and Holm [79,114] who propose that the electrical conductivity of carbon black depends on the separation distance between each particle [115] and the average size of each particle [116]. High pyrolysis temperatures are also known to give increased electrical conductivity to the pyrolyzed chars, recalcitrance, and tensile strength [117], explaining the high values of derived from EC vineyard pruning biochars and HCs prepared at 900 °C (Figure 12A) compared to those of almond shell and olive wood biochars prepared at 600 °C [9,94]. This finding is also aligned with the fact that using pyrolysis of vineyard pruning residues without HTC pre-treatment seems beneficial for obtaining lower EC values, and thus more disordered carbon materials. 

In the case of EDLCs, EC is of mayor importance, since the inner resistance of the electrode materials influence the overall performance and capacity of the EDLC [20]. That is why PHCs are preferred as electrode materials compared to hydrochars. As discussed in Section 4.3, the pre-step of HTC has an important influence on the formation of aromatic carbon structures, and therefore leads to higher EC values compared to the biochars obtained via pyrolysis of biomass without HTC. 

## 5. Conclusions

Returning to the initial research questions mentioned in the introduction, it was found that for the discussed applications some carbon properties correlate well (e.g., SSA should be as high as possible in both applications), whereas the importance of other carbon properties, such as EC, depends on the respective application. While for EDCLs, a high EC is of major importance, for certain fuel cell types, such as SOFCs or hybrid SOFC/MCFCs, high EC values are not necessary.

Furthermore, it was observed that comparing model substance and real biomasses, certain processes take place in both substances, e.g., the formation of better macro- and mesoporosity at the expense of microporosity when HTC is applied before pyrolysis. On the other hand, for cellulose, EC and SSA decreased with higher HTC temperatures and reaction times, whereas for pruning, lower HTC temperatures and longer reaction times led to better EC and SSA values and effects, which are associated with the presence of compounds such as lignin in the real biomass. Lignin is also assumed to be the reason for the worse C-balance of the PHCs from biomasses compared to those from cellulose.

In conclusion, it can be stated that biomass-derived carbon materials for the application in energy storage and conversion devices can be produced via pyrolysis, HTC, or a combination of both. Depending on the required properties of each application, reaction pathways and conditions have to be adapted individually. Whereas for DCFC applications, lower pyrolysis temperatures are recommended without a pre-treatment with HTC, materials for the application in EDLCs show the best properties if produced via high-temperature pyrolysis of HCs. 

In order to fully assess the obtained materials, testing of the most suitable materials in the respective application is required as a next step.

## Figures and Tables

**Figure 1 materials-12-01703-f001:**
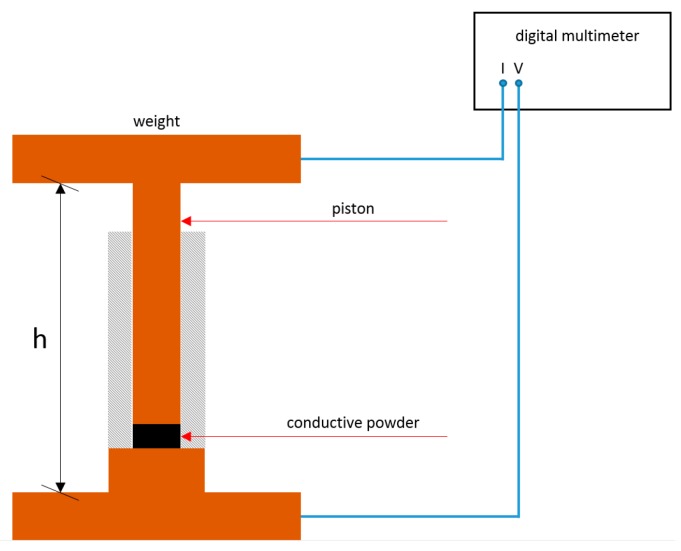
Scheme of the experimental device for the conductivity measurement.

**Figure 2 materials-12-01703-f002:**
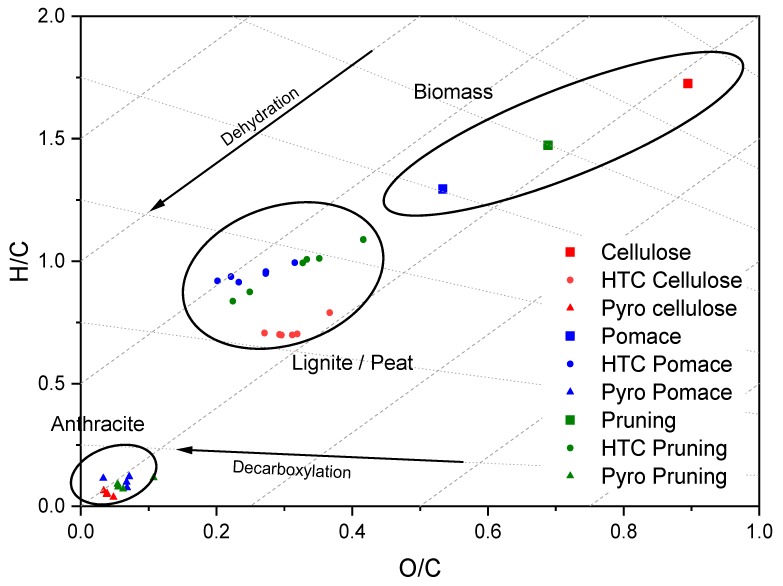
The van Krevelen diagram of different feedstocks and the resulting HCs (here HTC) and biochars (here Pyro).

**Figure 3 materials-12-01703-f003:**
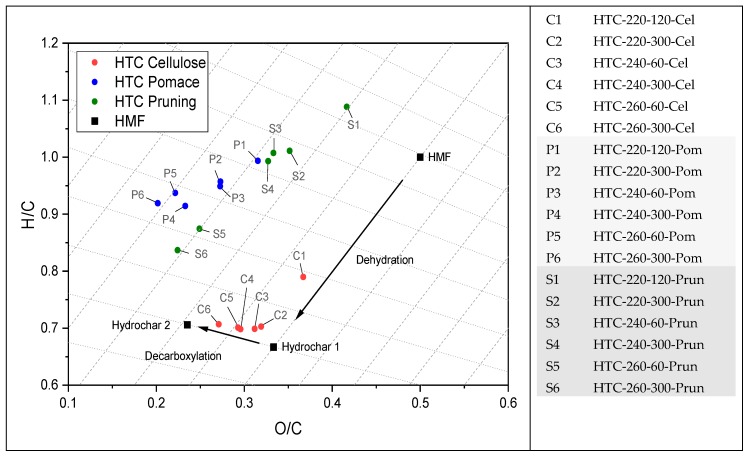
The van Krevelen diagram of HCs from different precursors. For comparison purposes, HMF and the hypothetical resulting HC after separation of H_2_O (Hydrochar 1) and subsequent separation of CO_2_ (Hydrochar 2) are also represented.

**Figure 4 materials-12-01703-f004:**
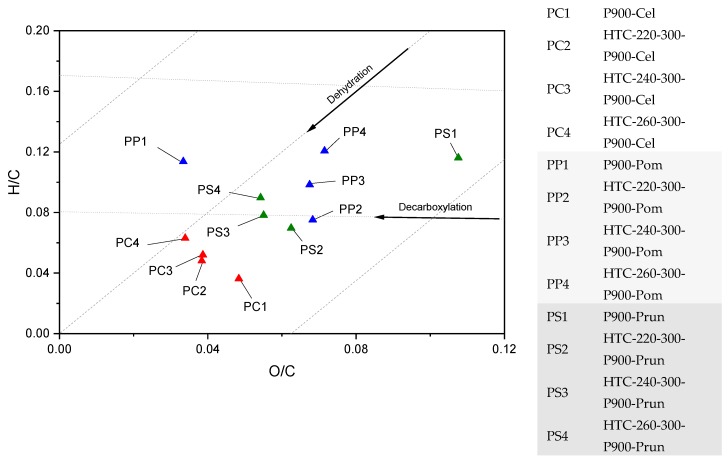
The van Krevelen diagram of BCs (PC1, PP1, PS1) from different precursors and corresponding PHCs (PC2-PC4, PP2-PP4, PS2-PS4).

**Figure 5 materials-12-01703-f005:**
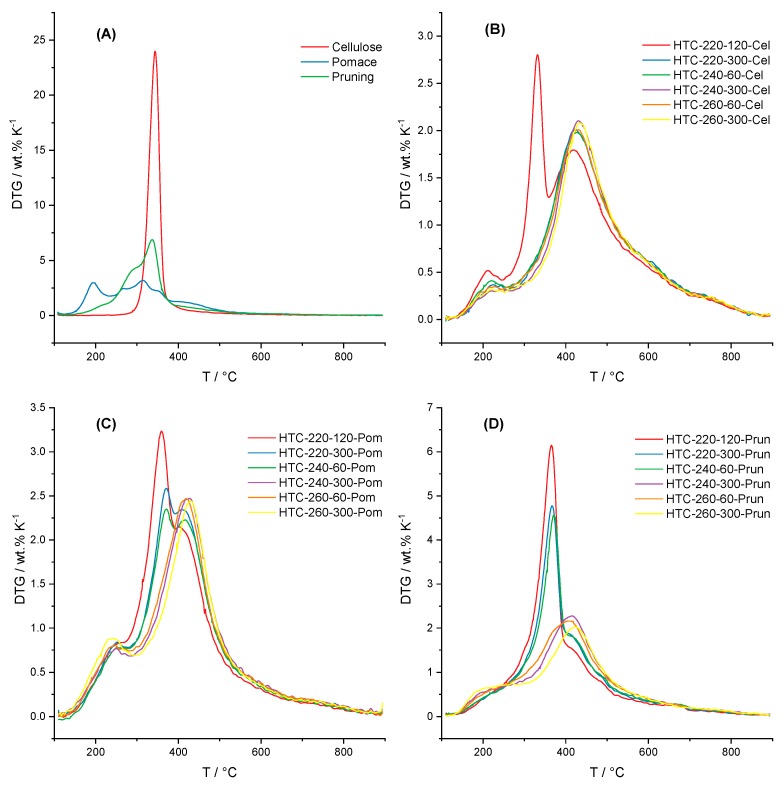
DTG curves of (**A**) feedstock and HCs from (**B**) cellulose, (**C**) pomace, and (**D**) pruning.

**Figure 6 materials-12-01703-f006:**
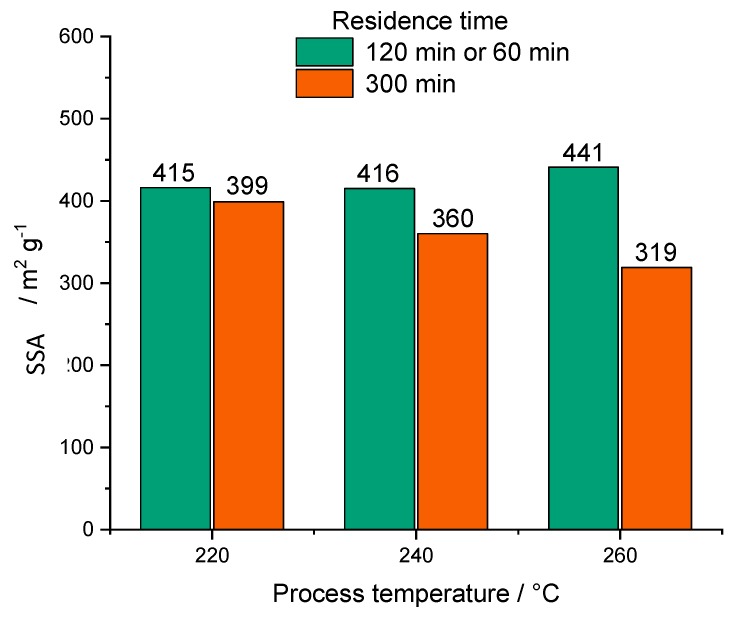
Specific Surface areas (SSA) (N_2_) of PHCs from cellulose vs. HTC temperature and reaction time.

**Figure 7 materials-12-01703-f007:**
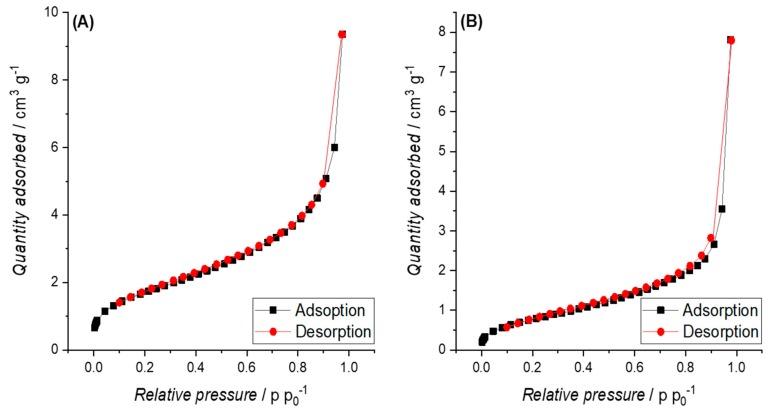
Nitrogen adsorption/desorption isotherms for (**A**) HTC-240-300-Cel and (**B**) HTC-240-300-Prun.

**Figure 8 materials-12-01703-f008:**
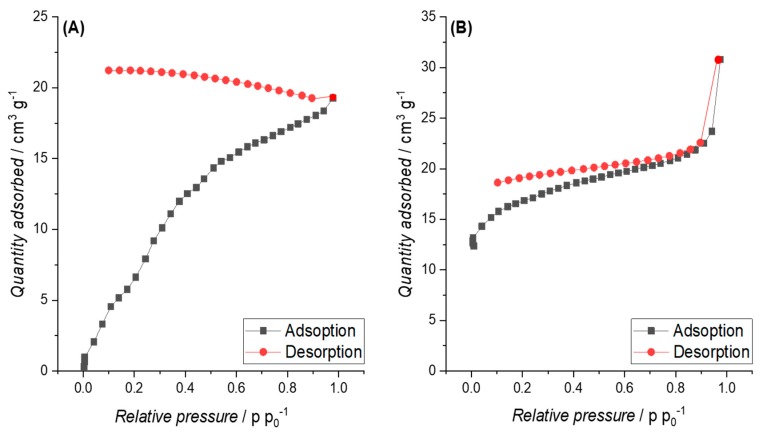
Nitrogen adsorption/desorption isotherms of (**A**) P900-Cel and (**B**) HTC-240-60-P900-Cel, representative for all pre-treated biochars from cellulose.

**Figure 9 materials-12-01703-f009:**
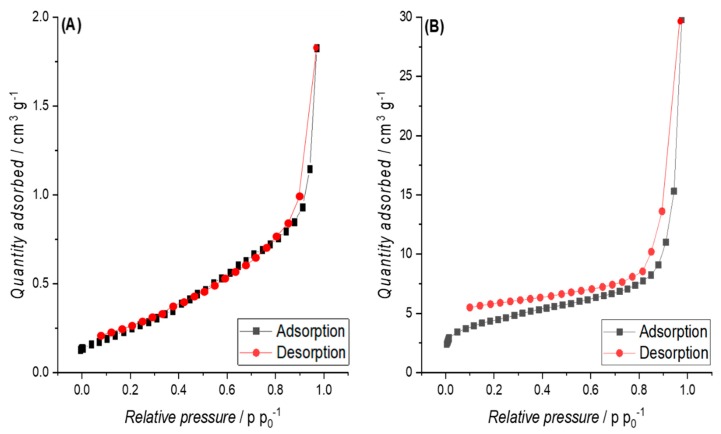
Nitrogen adsorption/desorption isotherms of (**A**) P900-Prun and (**B**) HTC-240-60-P900-Prun, representative for all pre-treated biochars from pruning.

**Figure 10 materials-12-01703-f010:**
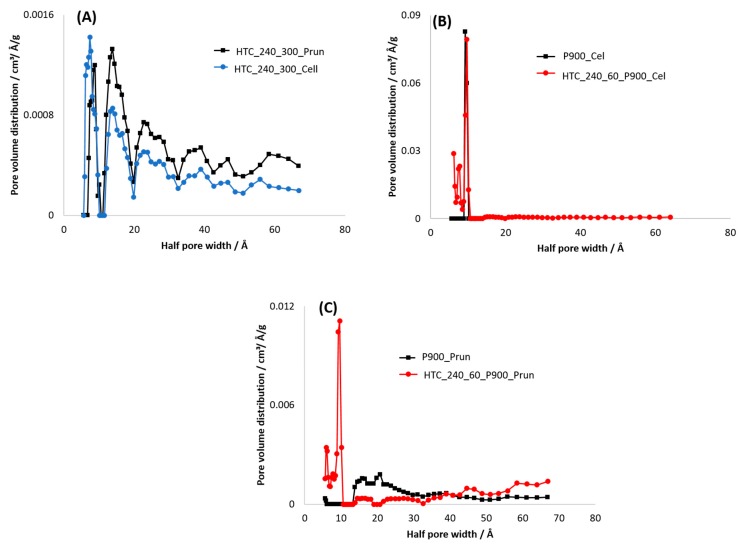
Pore size distributions for (**A**) HCs from cellulose and pruning, (**B**) BC and PHC from cellulose, and (**C**) BC and PHC from pruning.

**Figure 11 materials-12-01703-f011:**
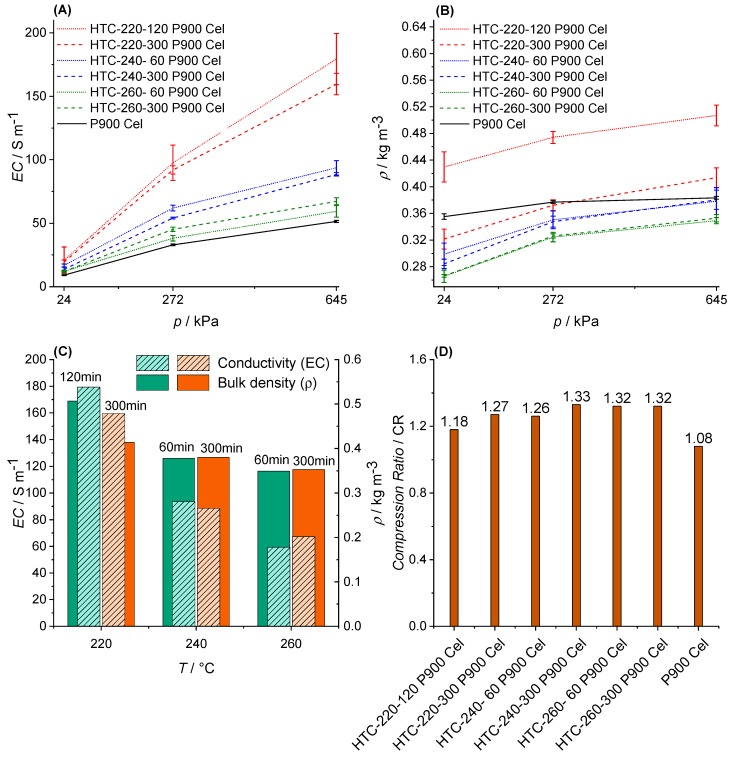
Comparative diagrams for BCs from cellulose and corresponding PHCs (error bars in (**A**) and (**B**) representing the standard deviation of the respective EC values) (**A**) EC vs. p, (**B**) ρ vs. p, (**C**) EC and ρ at 645 kPa vs. HTC operation temperature and reaction time, and (**D**) CRs.

**Figure 12 materials-12-01703-f012:**
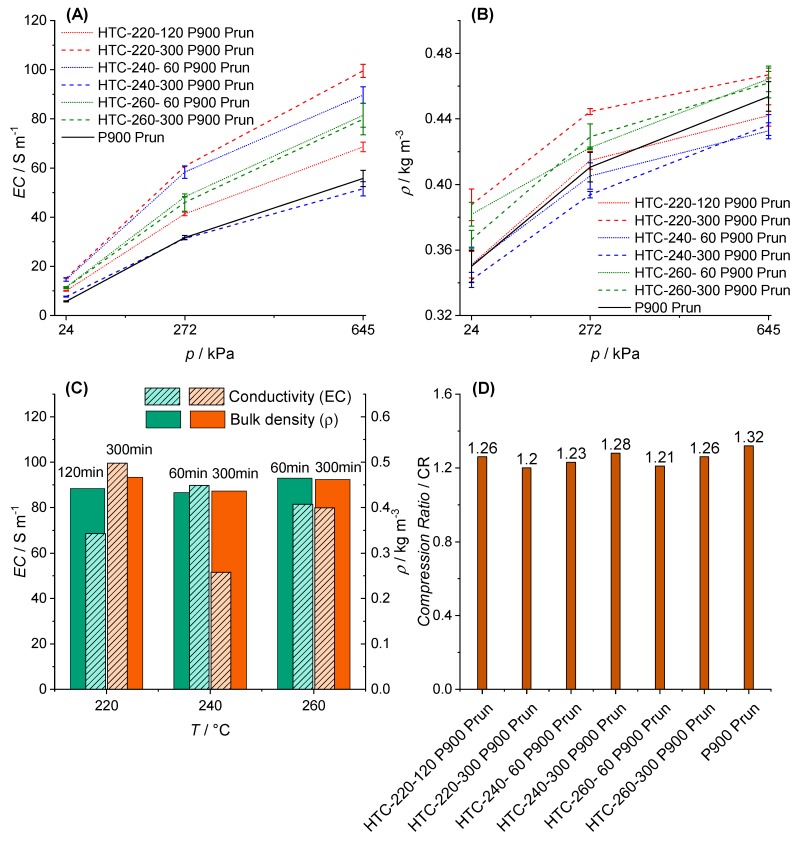
Comparative diagrams for BC from pruning and corresponding PHCs (**A**) EC vs. p, (**B**) ρ vs. p, (**C**) EC and ρ vs. HTC operation temperature and reaction time.

**Figure 13 materials-12-01703-f013:**
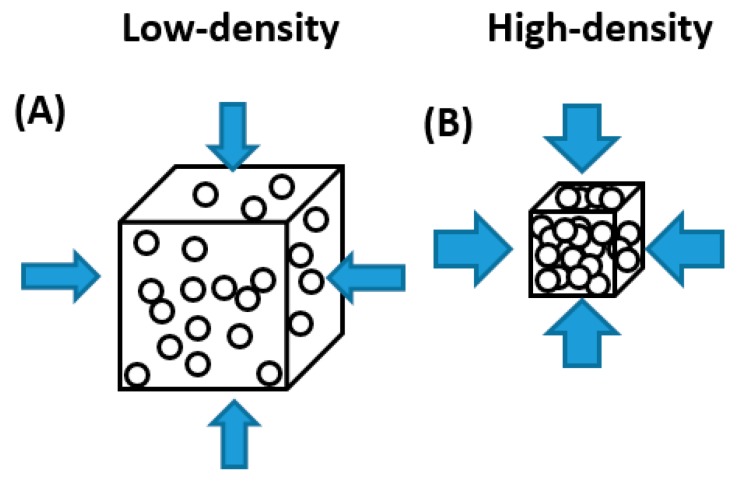
Schematized effect of compression on packing. (**A**) Low density, low compression, and (**B**) high density, high compression. The blue arrows represent the applied pressure p, the withe spheres represent the carbon particles.

**Figure 14 materials-12-01703-f014:**
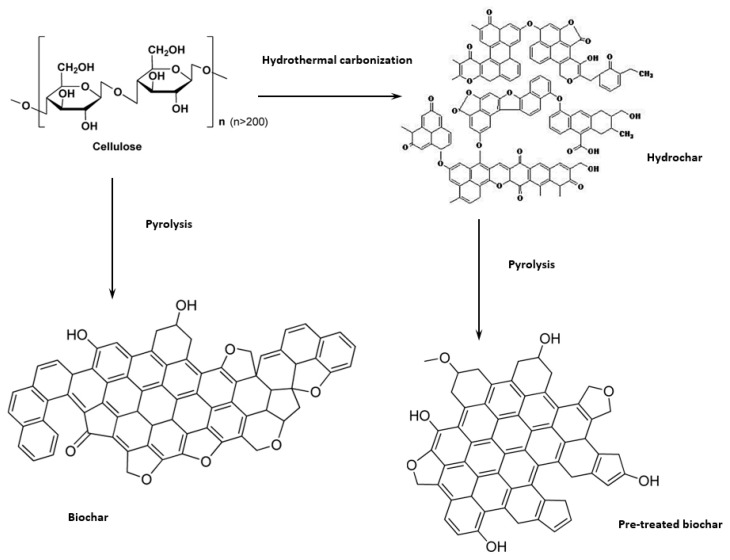
Production pathways of HCs (hydrochar), BCs (biochar), and PHCs (here pre-treated biochar) and the obtained schematic structures for cellulose as precursor (adapted from previous studies [67,68])

**Figure 15 materials-12-01703-f015:**
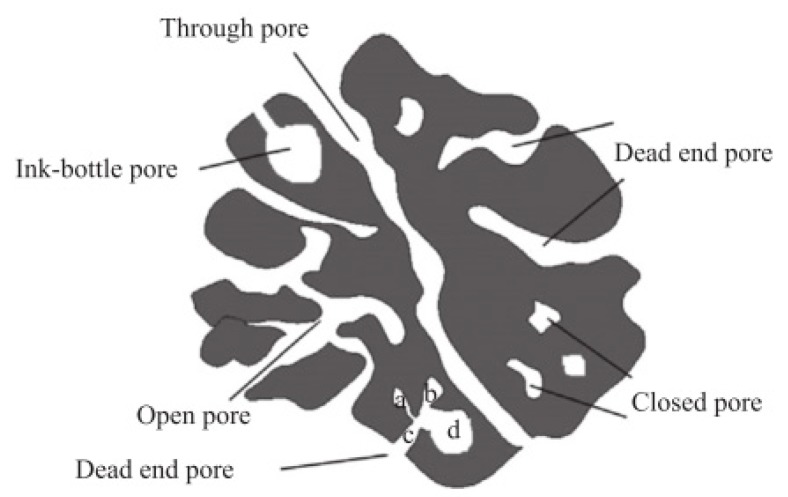
Possible pore types in Hydro- and Biochars, including (a) ultra-micro-, (b) micro-, (c) meso-, and (d) macropores (classified based on pore diameters (a) and (b) <2 nm; (c) 2–50 nm; (d) >50 nm) [82,89].

**Figure 16 materials-12-01703-f016:**
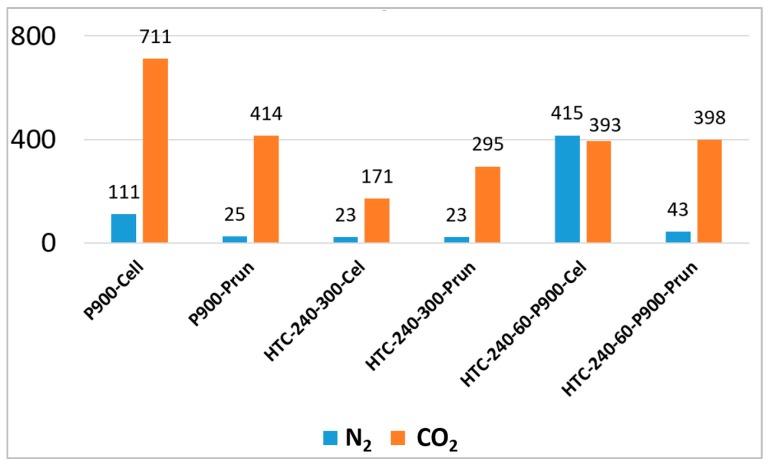
BET SSA measured with CO_2_ or N_2_ for selected materials.

**Table 1 materials-12-01703-t001:** Process parameters of hydrothermal carbonization for the production of hydrochars.

Process Temperature	Reaction Time
220 °C	120 min
	300 min
240 °C	60 min
	300 min
260 °C	60 min
	300 min

**Table 2 materials-12-01703-t002:** Proximate and elemental analysis of HCs and BCs and the resulting carbon yield from hydrothermal and pyrolysis treatment of cellulose, pomace, and pruning.

Sample Type	HTC		Pyrolysis (°C)	Elemental Analysis (Dry Basis, Ash Free) (wt.%)				Char Yield (wt.%)	C-YieldTotal ** (%)	pH Process Water
	Temperature (°C)	Time (min)		N	C	H	O			
BM	Cellulose			0	42.8	6.2	51.1		-	
HC	220	120		0	64.3	4.2	31.5	45	67.8	1.8
HC	220	300		0	67.4	4.0	28.7	45	70.9	1.9
HC	240	60		0	67.9	4.0	28.2	45	71.4	1.8
HC	240	300		0	68.8	4.0	27.2	46	73.3	2.2
HC	260	60		0	69.0	4.0	27.0	46	73.7	2.0
HC	260	300		0	70.4	4.2	25.4	45	74.5	2.2
BC	Cellulose *		900	0	94.9	0.5	4.6	24	52.1	
PHC	220	300	900	0	94.8	0.4	4.9	25	55.2	
PHC	240	300	900	0	94.7	0.4	4.9	26	57.5	
PHC	260	300	900	0	95.2	0.5	4.3	27	59.4	
BM	Pomace			1.8	54.0	5.8	38.4		-	
HC	220	120		1.7	65.4	5.4	27.5	56	68.0	4.1
HC	220	300		2.1	67.8	5.4	24.7	52	65.9	4.4
HC	240	60		2.0	67.9	5.4	24.7	52	65.3	4.4
HC	240	300		2.2	70.6	5.4	21.9	49	64.6	4.9
HC	260	60		2.2	71.2	5.6	21.1	49	64.0	4.9
HC	260	300		2.3	72.6	5.6	19.6	47	62.7	5.1
BC	Pomace *		900	2.3	92.6	0.8	4.2	33	56.2	
PHC	220	300	900	2.5	88.8	0.7	7.9	26	42.4	
PHC	240	300	900	2.7	88.7	0.5	8.1	26	42.6	
PHC	260	300	900	2.7	88.0	0.9	8.4	26	41.8	
BM	Pruning			0.8	48.6	6.0	44.6		-	
HC	220	120		1.0	60.2	5.5	33.4	61	75.8	3.7
HC	220	300		1.0	63.7	5.4	29.9	55	71.9	3.7
HC	240	60		1.2	64.7	5.4	28.8	52	69.0	3.7
HC	240	300		1.1	65.1	5.4	28.4	47	62.3	3.8
HC	260	60		1.1	70.3	5.1	23.3	44	64.2	3.7
HC	260	300		1.4	72.1	5.0	21.5	43	64.2	3.9
BC	Pruning *		900	1.4	85.5	0.9	12.3	28	49.3	
PHC	220	300	900	1.4	90.6	0.5	7.6	25	47.4	
PHC	240	300	900	1.5	91.3	0.7	6.6	25	47.5	
PHC	260	300	900	1.4	91.2	0.6	6.7	25	46.3	

Note: BM = Biomass; * only pyrolysis; ** only HTC or pre-HTC + pyrolysis.

**Table 3 materials-12-01703-t003:** Proximate analysis of feedstocks, HCs, and BCs. Values are given in wt.%. Ash content (AC) of cellulose is equal to 0. Note: FC = 100−Ash−VM.

Sample Type	HTC		Pyrolysis	Cellulose		Pomace			Pruning		
	Temperature (°C)	Time (min)	Temperature (°C)	FC	VM	FC	VM	AC	FC	VM	AC
BM *	Feedstock			24	77	33	67	5.1	28	72	2.7
HC	220	120	-	51	49	47	53	1.5	42	58	1.7
HC	220	300	-	55	45	49	51	1.5	46	54	1.8
HC	240	60	-	55	45	50	50	1.7	47	53	1.9
HC	240	300	-	57	43	52	48	1.8	54	46	1.9
HC	260	60	-	57	43	53	47	1.8	54	46	1.9
HC	260	300	-	59	41	55	45	1.8	57	43	2.0
BC	Feedstock		900	100	0	84	0	15.7	90	0	9.6
PHC	220	300	900	100	0	97	0	3.1	96	0	3.8
PHC	240	300	900	100	0	97	0	3.4	97	0	3.5
PHC	260	300	900	100	0	97	0	3.3	97	0	3.4

* BM = Biomass.

**Table 4 materials-12-01703-t004:** EC and ρ at 645 kPa of cellulose and pruning samples. n.d. = not determined.

Sample	EC (S m^−1^)	ρ (kg m^−3^)
Cellulose	0	n.d.
HTC-220-120-Cel	0	0.54
HTC-220-300-Cel	0	0.43
HTC-240-60-Cel	0	0.43
HTC-240-300-Cel	0	0.42
HTC-260-60-Cel	0	0.41
HTC-260-300-Cel	0	0.42
P900-Cel	51	0.38
HTC-220-120-P900-Cel	179	0.51
HTC-220-300-P900-Cel	160	0.41
HTC-240-60-P900-Cel	94	0.38
HTC-240-300-P900-Cel	88	0.38
HTC-260-60-P900-Cel	59	0.35
HTC-260-300-P900-Cel	67	0.35
Pruning	0	n.d.
HTC-220-120-Prun	0	0.48
HTC-220-300-Prun	0	0.48
HTC-240-60-Prun	0	0.49
HTC-240-300-Prun	0	0.48
HTC-260-60-Prun	0	0.50
HTC-260-300-Prun	0	0.49
P900-Prun	56	0.45
HTC-220-120-P900-Prun	69	0.44
HTC-220-300-P900-Prun	100	0.47
HTC-240-60-P900-Prun	90	0.43
HTC-240-300-P900-Prun	52	0.44
HTC-260-60-P900-Prun	81	0.46
HTC-260-300-P900-Prun	80	0.46

**Table 5 materials-12-01703-t005:** BET SSA of cellulose and pruning samples measured with N_2_ (normal letters) and N_2_ and CO_2_ (bold letters). n.d. = not determined; BM = Biomass.

Sample Type	Adsorbate	CO_2_	N_2_
	BET SSA	(m^2^ g^−1^)	(m^2^ g^−1^)
BM	Cellulose	**n.d.**	~1.1 [43]
HC	HTC-240-60-Cel	**n.d.**	19
HC	HTC-240-300-Cel	**171**	23
BC	P900-Cel	**711**	111
PHC	HTC-220-120-P900-Cel	**n.d.**	416
PHC	HTC-220-300-P900-Cel	**n.d.**	399
PHC	HTC-240-60-P900-Cel	**393**	415
PHC	HTC-240-300-P900-Cel	**n.d.**	360
PHC	HTC-260-60-P900-Cel	**n.d.**	441
PHC	HTC-260-300-P900-Cel	**n.d.**	319
BM	Pruning	**n.d.**	~1
HC	HTC-240-60-Prun	**n.d.**	22
HC	HTC-240-300-Prun	**295**	23
BC	P900-Prun	**414**	25
PHC	HTC-240-60-P900-Prun	**398**	43
PHC	HTC-240-300-P900-Prun	**n.d.**	73

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
