# Peer review of "Conductive Carbon Materials from the Hydrothermal Carbonization of Vineyard Residues for the Application in Electrochemical Double-Layer Capacitors (EDLCs) and Direct Carbon Fuel Cells (DCFCs)"

_materials, 2019, doi:10.3390/ma12101703_

Round 1

Reviewer 1 Report

Summary and general comment:

The devices for energy storage and conversion attract lots of research and application attentions recently. This contribution reports a bio-based carbon material for energy applications. Three char categories are systematically studied under various preparation conditions for understanding their effect toward energy storage device (electrochemical double-layer capacitors) and conversion device (direct carbon fuel cells). A few comments below are listed for the authors as a reference.

Additional Comments:

1. The data are well organized and analyzed. The reference list is amazing. Over 100 papers are cited in this paper. The reviewer apologizes for not being able to go over all these citations. Please check again and ensure the references are cited correctly.

2. In Figures 3.6 - 3.8, the adsorption and desorption isotherms should connect together at 1.0 P/P0.

3. Nitrogen adsorption/desorption isotherms should not have desorption isotherm lower than the adsorption isotherm. Please check again the data shown in Figure 3.8 A.

4. What are the pore sizes of the carbon materials? What would the pore sizes and pore-size distributions affect the device performances? The PSD analysis requested here would support the analysis in Figure 4.4.

5. Please correct the reference list and provide all references with sufficient publication information.

Author Response

Dear Sir or Madam,

Thank you for your comments on our paper, to which I will briefly refer in the following.

1. Thank you again. The references and citations were checked and adapted if necessary.

2. In Figures 3.6 - 3.8, the adsorption and desorption isotherms are connecting now at 1.0 P/P0, the missing points are displayed in the new graphs.

3. It is right, the desorption isotherm should not be below adsorption isotherm. The data shown in Figure 3.8 A was reviewed again and the graph was adapted.

4. The PSD analysis was added as Fig. 3.9. The following figures of chapter 3 were renamed and adapted in the text (former Fig. 3.9. is now Fig. 3.10., former Fig. 3.10 is now Fig. 3.11). The PSD results were briefly described in sector 3.3.2 (line 388-391) and discussed in sector 4.3 (line 643-705).

5. The reference list was corrected and missing publication information was added.

Please find attached the adapted manuscript.

Kind regards

Viola Hoffmann

Reviewer 2 Report

In this paper the authors  report the preparation and characterization of different biobased carbon materials which were shown to be applicable to energy storage and conversion devices, as it was also analysed for  EDLC and DCFC systems. In the introduction, the authors justify their interest in the selected subject,beginning to refer potential challenges concerned the properties of the carbons used as fuels which may affect negatively the expected good performances of the DCFC and EDLC systems. In this context they express the role of the SSAs and ECs of the electrodes, and correlate potential issues at  EDLCs and DCFCs. To tailor the carbon properties  they refer several procedures and focus their attention on the importance of vineyard  residues and their possibility to be converted into value-added precursors for thermochemical and other processes. Recent and appropriate 24  references were reported to enable the reader to understand more easily how they thought it would be possible to increase the performance of EDLCs and DCFCs  by means of adequate carbonizations of biomaterials leading to highly active electrodes and fuels for EDLCs and DCFCs, respectively. By the end of the introduction, the authors report that the focus will be on MCFCs and SOFCs, referred as DCFCs, but it is necessary to explain with some detail the relation between MCFCs/SOFCs and DCFCs, otherwise the readers will be confused with so many prepared  chars and assessed  devices (EDLCs, MCFCs, SOFCs).The authors should also correct reference 10 that should be written < SPS Badwal and S Giddey, The holy grail of carbon combustion --the direct carbon fuel cell technology. Materials Forum, vol 34, pp 181-185,2010>. In the Materials and Methods section, the production of hydrochars, biochars, and pyrolysed biochars, is shortly but adequately  described. This section also included chars analysis, namely TGA , proximate and elemental analyses, carbon yield, and electric conductivity and bulk density measurements. In the third section on results, the thermochemical conversion data was listed in 2 tables and illustrated in 3 figures with van Krevelen diagrams, being critically and usefully analysed. The carbon decomposition behaviour of the precursors and HCs led to DTG curves, evidencing  characteristic sugars and tannins peaks. Further tables showed ECs and bulk densities as well as SSAs of the different precursors. Physisorption isotherms gave further information on the chars. Nearly all tables and figures were deeply discussed either in scientific terms or as a result  of practical observations, so a piece of excellent work really very well structured and informative. In the discussion section, the authors maintained the previous well-structured , scientifically deep and clear approach regarding the thermochemical conversion, the thermal decomposition, and EC and physicochemical properties. The section finalizes with potential applications of the prepared and characrterized samples to EDLCs and DCFCs. The reported critical assessment is excellent but as the authors express at the end of the conclusions section , testing of the most suitable materials in capacitors and fuel cells is required as a next step . The article is excellent requiring publication.

Author Response

Dear Sir or Madam,

Thank you very much for your comments on our paper, to which I would like to refer in the following.

The relation between SOFCs/MOFCs and DCFCs is specified in line 139-142.

Reference 10 was adapted.

Please find attached the adapted manuscript.

Kind regards

Viola Hoffmann